# Sustainable Supply Chains: Evidence from Small and Medium-Sized Manufacturers

**Jun-Der Leu \*, Larry Jung-Hsing Lee, Yi-Wei Huang and Chia-Chi Huang**

Department of Business Administration, National Central University, No. 300, Zhongda Rd., Zhongli District, Taoyuan City 320317, Taiwan; larrylee@mgt.ncu.edu.tw (L.J.-H.L.); hiw@ey.gov.tw (Y.-W.H.); leu1431@yahoo.com.tw (C.-C.H.)

\* Correspondence: leujunder@mgt.ncu.edu.tw; Tel.: +886-03-4227151 (ext. 66123)

**Abstract:** As a result of the 1997 Kyoto Protocol, numerous countries have become highly interested in sustainability. Manufacturers have been triggered to develop sustainable supply chain strategies in consideration of their value-added processes and stakeholders. This study was conducted, in the context of small- and medium-sized enterprises in the Taiwanese manufacturing industry, in order to examine the relationship between sustainability and business performance from the perspective of the supply chain aspects of business cooperation, operation integration, and enterprise commitment. A total of 290 companies were surveyed and structural equation modeling was applied to analyze the figures from the samples. The results reveal that the implementation of sustainable practices in the context of enterprise commitments, supply chain cooperation, and operation integration has a positive impact on sustainability and business performance. Furthermore, in high-value manufacturing, supply chain cooperation significantly affects sustainable business performance through internal operation integration. These results may serve as a reference for the realization of supply chain sustainability in small- and medium-sized enterprises.

**Keywords:** small- and medium-sized enterprise; supply chains; fit; structural equation modeling; sustainable supply chain; sustainable development



## 1. Introduction

Sustainable development (SD) has become a primary objective in enterprise strategy systems, and is achieved via the integration and balancing of the three dimensions of sustainable development: economic, social, and environmental [1]. Increasingly complex global-scale production, transportation networks, and value chains extend SD to entire supply chains. All the partners in a network, not only a single company, must confront future environmental challenges. The manufacturing industry accounts for 30% of Taiwan's GDP and encompasses industries, such as IC and computer products, that are exported worldwide [2,3]. The manufacturing industry is the driving force of Taiwan's economy, aiding its ability to meet stakeholder expectations of environmental regulations and compliance. It is critical for manufacturers to develop a high-value manufacturing strategy to ensure the sustainability of the national economy. The concept of sustainable supply chain management is very important to the business strategies of small- and medium-sized enterprises [4]. Companies can enhance added value through numerous methods, among which one of the most crucial is the enhancement of both their corporate image and customer loyalty by the demonstration of a contribution to society through products, services, or local care [5].

Most enterprises have recognized the importance of SD, but ambiguity about the methods of its implementation and its impact on corporate performance has hampered applications by various enterprises to establish long-term SD plans. Originally, supply chain management was perceived as process-oriented and customer-focused, with material flowing from upstream suppliers to downstream customers [6,7]. Currently, companies are

developing supply chain management strategies in order to boost supply chain cooperation in response to market changes and complexities [8]. Supply chain management has had a positive and significant impact on sustainable performance [9]. Therefore, most enterprises have a practical understanding of how to establish internal SD operations from an environmental perspective, including production process improvement, green procurement, and product greening.

SD is a long-term strategic objective to sustain business continuity without impacting the ability to meet the needs of future generations [10]. Enterprises may contribute to SD through three steps: the first stage is to address their internal operations for achieving sustainability; the second stage is to develop an SD strategy, in particular, the integration of external supply chains that are conducive to SD; and the third stage is the extension of sustainable issues outside of an enterprise, something that requires an explicit commitment to social responsibility, as well as environmental and economic factors [11]. Therefore, this study aimed to address the issue of how to increase the sustainability of supply chain cooperation and performance through the expansion of sustainable programs or activities in a company's internal operations. Although numerous topics can be evaluated through qualitative research [12–16], such research may be limited by the sample size. Therefore, this study used quantitative research methods to explore covariation and fitting model patterns as an intermediary in sustainability supply chain (SSC) relationships.

This study explored the realization of SSCs in small- and medium-sized enterprises. First, a literature review was conducted to identify the sustainable operation practices of enterprises in the context of SSCs and to develop a research framework. Second, the research framework was employed to develop questionnaires and hypotheses. Finally, the structural equation modeling was applied to analyze the research samples. Questionnaires were administered to test the fit in terms of covariation and mediation and to identify the consistency of various facets, as well as to determine the relationship between sustainable performance and business performance. The contributions of previous authors are list in Table 1. Therefore, this study developed the concept model of SEM, which is shown in Figure 1.

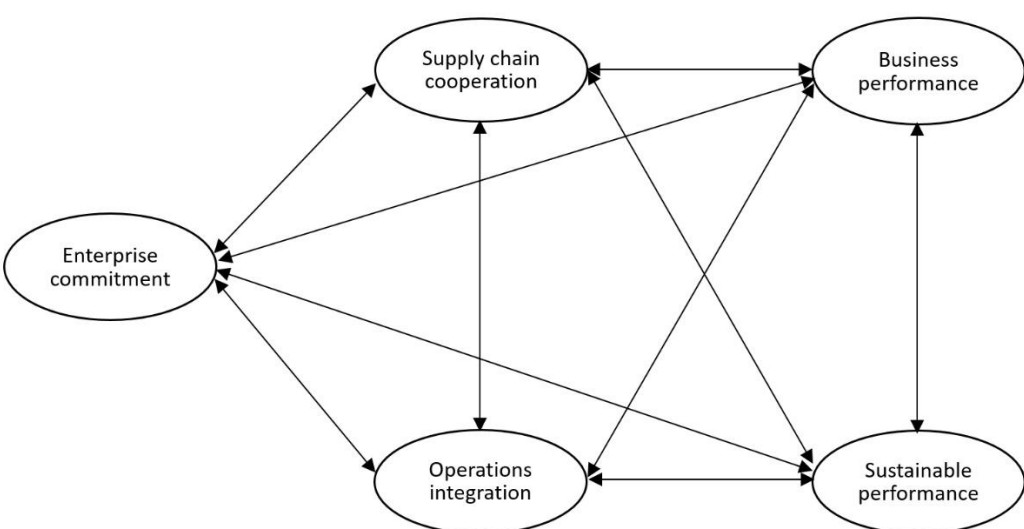

**Figure 1.** The concept model of SEM.

Section 2 presents a literature review investigating high-value manufacturing, enterprise and sustainability, SSCs, and performance. Section 3 describes the questionnaires developed, sampling, and methods used. Section 4 analyzes the results through the relevant aspects by using SEM. Section 5 presents the conclusions, the academic as well as management implications, and recommendations for future research.

**Table 1.** The contributions of previous authors.

| Author(s) | Sustainability | Enterprise | Manufacturing | Supply Chain | Methods |
|---|---|---|---|---|---|
| Desa [1] | Include | Include | Include | Include | NA |
| Chow et al. [2] | NA | Include | Include | Include | SEM |
| Koh et al. [3] | Part | Part | Part | Green | Case |
| Kot [4] | Include | Include | Include | Include | Survey analysis |
| Martineau [5] | NA | NA | NA | Reverse logistics | OR |
| Fleischmann et al. [6] | Recycle | NA | Part | Reverse logistics | Review |
| Cooper et al. [7] | NA | Part | NA | SCM | Review |
| Paulraj and de Jong [8] | ISO-14001 | Include | NA | NA | Event-study |
| Li et al. [9] | Include | Part | Part | SCM | SEM |
| WCED [10] | Include | Include | Include | Include | NA |
| van Marrewijk [11] | Include | Include | NA | NA | Review |
| Jayaram and Avittathur [12] | Include | Include | Include | Include | Review |
| Paliwal et al. [13] | Include | Include | NA | Block chain | Review |
| Bui et al. [14] | Include | Include | NA | SSC | Data-driven literature review |
| Khan et al. [15] | Sustainability-related risks | Planning-related risks | NA | Logistic and outsourcing related risks | Fuzzy Delphi and fuzzy DEMATEL |
| Signori [16] | Include | Include | NA | SSC | Theory-building grounded |
| This study | Include | Include | Include | Include | SEM Mediation fit model |

## 2. Literature Review

This section defines the concepts of high-value manufacturing, enterprise commitment and sustainability, SSCs, and performance on the basis of a literature review.

### 2.1. High-Value Manufacturing

In economics, added value is defined as the value that an enterprise creates through production or manufacturing or, more specifically, the difference between the output and the intermediate input costs. Price is no longer the main competitive advantage in competition between firms. Instead, an enterprise's capacity to innovate, optimize processes, or sustainably deliver value to stakeholders is prioritized [17]. As manufacturing companies grow, they tend to move toward high added value [18]. Specifically, they implement more complex operating activities, at which point competitive advantage is no longer achieved solely through pricing, but instead by transferring value across multiple operations to create a broader, more balanced, complex portfolio of activities.

Numerous business owners have stated that, although price has become a health factor, manufacturing processes have not necessarily lost their importance. Companies will continue to focus on core production processes and support them through other services or design activities [19]. When companies can successfully integrate intangible activities such as services and ideas with tangible operational activities, they can create opportunities for new competitive advantages. High-value enterprises widely apply this logic when combining products and services to achieve added value in various departments. Consequently, service cost, which was once considered a major resource to avoid, has been transformed into a critical resource that increases advantages and prevents manufacturing profits from declining in numerous developing countries [20].

### 2.2. Enterprise Commitment and Sustainability

In today's global economy, it is critical for companies to embrace social and environmental responsibility in order to meet the demands of their investors, consumers, employees, and communities [21]. Companies are globally answerable for the quality of their operations, which encompasses their actions regarding human rights, health, safety standards, and environmental performance [22]. Leaders determine the scope and degree of corporate social responsibility, and form corporate culture through cognition, norms and regulations which are then conveyed to employees and customers [23,24].

When enterprises are willing to adopt higher international certification standards, representatives are willing to accept higher normative standards in the pursuit of long-term operations rather than short-term profits [25]. Therefore, positive results can be obtained in regard to internal processes, customers, and the management of suppliers. In macro institutional theory, whether industries can effectively self-regulate is dependent on the degree to which enterprises are willing to practice social responsibility. Therefore, if an enterprise can become a leading manufacturer in a supply chain on its own, it can influence industry norms to meet desired standards. In addition, leaders can transform partnerships in a supply chain into a sustainable collaboration through sustainable thinking and demonstrating a willingness to share resources to meet the sustainable goals of suppliers and customers through enterprise commitment [8]. Therefore, such a commitment represents a thorough understanding of the responsibilities and capabilities of the business, and its partners, in terms of sustainable management. Only when the entire supply chain has a shared goal can sustainable practices produce tangible profits.

### 2.3. SSCs and Performance

SSCs are a very important field that enterprises must work hard to develop [26]. Extending environmental sustainability practices throughout the supply chain is challenging because of the numerous uncertainties involved. "First, the lack of regulatory control over other firms in the supply chain prevents the full presentation of the risk-reduction benefits (e.g., fines or prohibitions on sales of products that do not comply with laws and regula-

tions) that the original enforcement environment can provide to the enterprise." SSCs only work if companies can confirm that upstream and downstream partners are committed to implementing sustainable practices and strive to avoid negative externalities [27–29].

A resource-based view can be used as the basis for the consideration of SSCs. The economic theory of natural resources incorporates environmental resources into resource-based theory, which considers the scarcity of resources [30]. However, such implementation requires technological development and SD teams to develop implicit knowledge, which is difficult to imitate [31]. The adoption of waste reduction is the most important environmental factor in SSCs, and has a vital impact on the financial and positive results of enterprises. The timely and legal payment of taxes and fees, provision of health and safety equipment, and applications of ethical business and trade standards are the most important factors [4]. Social complexity, however, involves advanced environmental management practices such as environmental design and life cycle analysis, which require the integration of various stakeholders in the supply chain [32]. Finally, proprietary attributes represent a resource that must be company-specific, and can be achieved through technological development or through the transformation of resource applications in emerging markets and developing countries [33]. This study mainly investigates the social complexity of central enterprises and supply chain partners and whether they can create company-specific assets.

Numerous studies have examined the correlation between sustainable collaboration, internal integration, supply chain integration, and performance; the integration of corresponding internal functions contributes to improvements in business performance [34]. Internal integration can help enterprises overcome functional barriers, which can stimulate cooperation through information sharing, joint planning, and collaborative integration; in this manner, the external requirements of enterprises can be met, thereby improving the efficiency of enterprise processes. The integration of external supply chains can help supply chain partners understand each another's needs. By exchanging products, processes, and capabilities, companies can be more efficient and resilient in developing production plans and manufacturing products. In addition, a close relationship with customers can enhance the accuracy of demand forecasting and reduce errors in design, timing, and inventories. External supply chain integration, whether it involves upstream suppliers or downstream customers, has a positive impact on business performance.

However, when enterprises and supply chains become integrated, the effectiveness of environmentally sustainable practices in business performance is unclear. Ultimately, the resources for implementing environmentally sustainable practices are an opportunity cost for other potential purposes that may be overlooked. Consequently, environmental sustainability has a significant negative impact on the financial performance of a business [35].

The effectiveness of environmentally sustainable practices in supply chains is uncertain. Environmental sustainability has a negative impact on performance in corporate procurement, and scholars have suggested that sustainable sourcing can only have a partially positive impact on the environmental, economic, and social performance of businesses [36,37]. Therefore, an enterprise's financial results should not be the only basis for evaluating performance. When enterprises in different industries and stages of development have distinct objectives, performance considerations should be based on various perspectives.

## 3. Research Model

Fabbe-Costes et al. [38] organized the scanning objectives of an enterprise into several levels, including the social, network, supply chain, and corporate levels, as well as corporate social responsibility and the commitment and attitude of a company's stakeholders (the human level). On this basis, and because of the various differences in sample characteristics, which focus more on products and processes but not the market, and research methods, the three major structures—namely, enterprise commitment, supply chain integration, and operational mechanism integration—are incorporated into the proposed model.

Secondly, a direct or indirect causal relationship exists among supply chains, sustainable production, and performance [39]. For the proposed model, a "sustainable performance" structure is implemented, and "market performance" is replaced with "business performance", primarily because of the differences identified in the study sample (Taiwan's manufacturing industry). Manufacturing in Taiwan is focused mainly on products, processes, and weak links in the market, which mostly comprise contracts with original equipment manufacturers and original design manufacturers. The incorporation of a market performance structure in the proposed model may reduce its credibility; therefore, it is replaced with manufacturing industry performance, which is associated with a business's performance structure.

Finally, the research confirms the progressive relationship between sustainability and economic benefits [40]. Therefore, refined production is discussed at the working environment and strategy levels, and the proposed model references research to divide the structure into general and strategic levels. Among them, the supply chain is considered to be part of the general level, and the operational mechanism is part of the strategic level because of its association with sustainability strategies. Different structural levels are used to determine the model's capability for influencing performance and identifying the differences in influence between levels.

The models used in this study were based on the three aforementioned studies. The subsequent section describes the relationship between the various structures, how the structures are measured, the research model, the research hypothesis, and the structures themselves.

### 3.1. Enterprise Commitment

Enterprise commitment refers to the business philosophy and strategic development plans of an enterprise. The concept of SD at a corporate level is described as corporate sustainability, and is based on the three pillars of economic, ecological and social issues [41]. Previously, corporate social responsibility (CSR) mainly referred to what companies were required to contribute socially and ethically [42,43]; however, globalization, SD, and other concerns have resulted in the extension of CSR to human rights, diversity, and the environment [44,45]. Enterprises increasingly feel internal and external pressures that encourage them to embrace sustainability and appropriate SD management [46]. This is also demonstrates how CSR, as a new tool, fits into the current corporate responsibility or sustainability framework in order to complete the picture of corporate sustainability [11]. Stakeholders have become the main factor influencing enterprises to incorporate environmental responsibility into their business policies. Moreover, when business philosophy no longer defines short-term profit as the main goal, enterprises begin to focus on the relevant social, environmental, and other long-term value-added indicators. Finally, companies that adopt high inspection standards, such as ISO4001, can demonstrate their commitment to environmental sustainability.

### 3.2. Supply Chain Cooperation

Supply chain cooperation refers to the degree of cooperation and the similarity of objectives between an enterprise and its upstream suppliers and downstream customers. Supply chain cooperation represents the integration of materials and information [47–49]. Material integration flows upstream to downstream, whereas information integration flows downstream to upstream [50]. By obtaining extensive information from trading partners, center manufacturers can accurately determine the status of inventory, transport, and production schedules, and immediately revise them in order to refine production processes. This integration with the downstream end also promotes customer satisfaction, which directly or indirectly affects customer relationship management through the flow of information. During material and information integration, the relationship between enterprises and the upstream and downstream ends gradually transitions from a short-term to a long-term relationship [51]. Central manufacturers may not maintain as many

partners or switch between as many contracts as before, but they will thereafter select who they cooperate with by evaluating the overall minimum cost of cooperation. Therefore, enterprises can enhance the evaluation ranking system of companies in the supply chain. Furthermore, a circular economy, which involves limiting the amount of raw material consumption that harms the environment, is exactly what economists expect in terms of sustainable economic growth. Botezat et al. [52] analyzed the practical applications of green supply chain cooperation and concluded that high- and low-level members of a green supply chain affect circular economy practices differently.

### 3.3. Operation Integration

Operation integration is required for enterprises to function properly. Procurement, production, logistics, sales, and manufacturing research and development are the focuses of companies in terms of transferring value [53]. In addition to the emergence of low-carbon production, green warehousing, and sustainable transportation, low-emission vehicles have been implemented on a large scale to increase transportation sustainability. Although cost and vehicle reliability are crucial factors in commercial vehicle procurement, the consideration of electric vehicles during the decision-making process is an indicator for legitimate investments in sustainability [54]. Sales are activities that motivate consumers to purchase, including advertising, promotions, access, and after-sales services. [55]. However, because of increasingly complex products and pressure from external stakeholders, enterprises must employ considerable care in sales. The scope of operation is no longer limited to product sales; end-of-life product management is now also a major consideration [29]. Product complexity is positively associated with the consumption of energy and environmental resources. The carbon footprint of a product's life cycle can indicate the extent of the product's impact on the environment. Extending product life cycles demonstrates environmental responsibility and enables enterprises to have greater control and supervision over a product, which further enables the production summary to clearly indicate responsibility at each stage of a product's life cycle.

### 3.4. Sustainable Performance

The impact of supply chain management should be determined based on three main aspects of sustainability: environmental performance, social responsibility and economic contribution [56,57]. Here, sustainable performance refers to the capability of an enterprise to manage the environmental hazards of its products. Companies prioritize environmental performance mostly because of regulations, contractual commitments, social perspectives, and competitive advantages [58]. In a circular economy, waste recovery and recycling are essential for effectively achieving sustainability goals. Therefore, the environmental impact of recycling should be carefully considered [59]. However, the public typically only comes into contact with final products. Thus, they pay greater attention to the product and the management of its life cycle. Other stakeholders typically focus on the product and its production, depending on their role and requirements. Production guidelines include the well-established 3Rs (reduce, reuse, and recycle) as well as the current concept of 6Rs (reduce, reuse, recycle, restore, redesign, and remanufacture). To ensure that process planning is of the highest quality, enterprises should focus on practices such as energy reduction, resource consumption, and toxic waste removal. Studies have also revealed that stronger connections within enterprises generate more sustainable performance [60,61].

### 3.5. Business Performance

Business performance is a measure of an enterprise's production capacity [62]. Companies with close partnerships in the supply chain can increase their competitive advantage, which indicates a positive relationship between their internal integration and business performance [63]. Indicators of competitive advantage and business performance include responsiveness to market changes, process efficiency, time, cost, and quality [64,65]. Therefore, this study divided production capacity into products and processes for evaluation

purposes. Under strict environmental monitoring, production is less likely to be interrupted, which positively affects the capacity and unit cost [66,67]. In terms of the product facet, customer satisfaction affects a product's acceptance in the market. Customer satisfaction is not easily quantifiable; therefore, the final product yield and delivery rate can be used as indicators [68].

### 3.6. Covariation Fit Model and Hypotheses

Covariation fit refers to the internal consistency between a set of related variables, and internal consistency refers to the concept of adaptation [69]. When a company commits to environmental sustainability, the company is willing to make relevant, long-term investments and exchange resources and knowledge, which promotes cooperation [70]. Enterprises share and use the complementary resources between them in order to produce and optimize products. Moreover, collaboration can generate benefits such as process efficiency, supply elasticity, business synergy, quality, and innovation. When employees perceive their company to be committed to implementing sustainability policies, they recognize the importance of perpetuity and implement it in their daily work. With rapid changes in the environment, the performance indicators adopted by enterprises should be more comprehensive. Such performance indicators should include both financial and operational indicators [64,65], and because environmental issues are not short-term objectives, the indicators that companies adopt should also involve products and processes that promote environmental sustainability [70]. Therefore, a covariation fit model is established, including dimensions such as corporate commitment, supply chain cooperation, operation integration, sustainable performance, and operational performance, as shown in Figure 2. Additionally, Hypothesis 1 (H1) and Hypothesis 2 (H2) are proposed as follows:

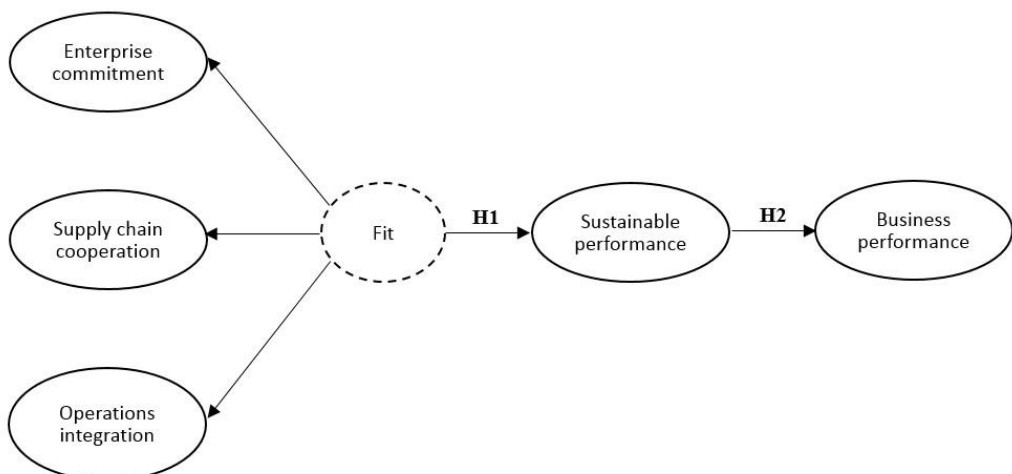

**Figure 2.** Covariation fit model for SSCs.

**Hypothesis 1 (H1).** *In a sustainable manufacturing supply chain, supply chain cooperation, enterprise commitment, and operation integration have a positive impact on sustainable performance.*

**Hypothesis 2 (H2).** *Sustainable performance has a positive impact on an enterprise's business performance.*

### 3.7. Mediation Fit Model and Hypothesis

Enterprise commitment involves management support, vision statements, departmental objectives, training, and evaluation [61]. The sustainability department can be seen as an inimitable resource. In the division, the skills and abilities of managers are integrated and reorganized to create new strategic value, which shows substantive empowerment towards sustainability [46]. The main task of senior management personnel is to combine environmental management with strategic leadership and organizational development

policies. After receiving support from senior executives, midlevel executives become the main decision-makers for accelerating the adoption of environmental management practices. As decision-makers, organization stakeholders and mangers must work together to overcome obstacles in order to achieve a sustainable and agile supply chain. [71]. When more midlevel management personnel promote sustainability practices that are more progressive than existing regulations or those of their competitors, they are more likely to implement SSCs [72]. Therefore, Hypothesis 3 (H3) is proposed as follows:

**Hypothesis 3 (H3).** *Enterprise commitment has a directly positive impact on sustainable performance.*

Generally, scholars agree that the most crucial benefit of cooperation in supply chain management is knowledge exchange and interorganizational learning [73]. Through the collection, integration, transformation, and creation of knowledge, enterprises can develop a stronger competitive advantage in the supply chain. No single enterprise can achieve this type of synergistic effect on its own. The competitive advantages gained from supply chain partnerships can help enterprises to increase profits, reduce purchase costs, add value, and improve performance in various manners [74,75]. Simatupang and Sridharan (2005) proposed the Co-op Index, which measures the degree of cooperation between enterprises. They identified a positive relationship between the Co-op Index and business performance [76].

When companies in a supply chain collaborate, they can gain a clearer understanding of each other's commitments and objectives toward sustainability [8]. Enterprises should set long-term sustainable development goals, report transparently, develop a sustainable development culture, and manage supply chain risks appropriately [77]. Organizations usually adopt sustainability practices not due to inside-out decisions, but rather because of various external pressures, such as from the government, competitors, and consumers [78]. However, these partnerships rely on reciprocity, and when companies work together on sustainability, they also expect their partners to commit to environmental matters. The aforementioned benefits or performance gains can only be obtained if consistent objectives are established throughout the supply chain. If one company considers reducing its investment in its partnerships, other companies may be incentivized to follow suit, ultimately affecting the performance of all the companies involved. Thus, enterprise commitment is the greatest driver of all behaviors related to sustainability practices. Therefore, hypotheses 4a and 4b (H4a and H4b) are proposed as follows:

**Hypothesis 4a (H4a).** *Enterprise commitment has a direct positive impact on sustainable performance through supply chain cooperation.*

**Hypothesis 4b (H4b).** *Enterprise commitment has a direct positive impact on business performance through supply chain cooperation.*

Sustainable practice is not a single-stage activity, but, rather, an activity that spans various value chain sectors and different stages of the enterprise life cycle [79]. Having an interdepartmental, cross-functional, and product-wide life cycle that considers whether the business has achieved its sustainable goals is essential. Conversely, if only a single department has the resources and knowledge to implement sustainable practices and does not share them with other departments, achieving sustainability goals will be challenging. Therefore, for an enterprise to achieve the most effective sustainability practices, variation in departments' sustainability goals must be minimized, but expansion of these departments' sustainability goals must be maximized. Enterprise commitment is an essential means for achieving such goals. Employees in diverse departments tend to have distinct perspectives on management practices. Not all departments consistently implement the management guidelines of the enterprise [80]. Therefore, the enterprise's commitment is essential to ensure that all employees throughout all departments feel a part of the enterprise's sustainable practices. Furthermore, most departments have competitive attitudes; therefore,

an enterprise must demonstrate its commitment by convincing departments to collaborate in pursuit of maximizing their benefits. Therefore, hypotheses 4c and 4d (H4c and H4d) are proposed as follows:

**Hypothesis 4c (H4c).** *Enterprise commitment has a positive impact on sustainable performance through operation integration.*

**Hypothesis 4d (H4d).** *Enterprise commitment has a positive impact on operating performance through operation integration.*

The mediation fit model can be used to determine whether a variable exhibits a mediation effect in a relationship. Although an enterprise may be part of a close supply chain network, improving performance through partners may not be possible if the enterprise's relationship with its partners is not effectively employed [39], and because the benefits to be gained from such relationships are often complex and rely on causally ambiguous knowledge [81]. Enterprises must have an absorption capacity sufficient for translating these relationships into benefits and performance improvements.

Business performance is essential for the sustainability of an enterprise. The successful implementation of environmentally sustainable practices provides an enterprise with the opportunity to improve its core competitiveness or added value [82,83]. The board's support and commitment to an enterprise's SD are also crucial for sustainable performance [84]. Florida [85] states that to promote more environmentally sustainable manufacturing, external suppliers and customers must be integrated with internal management strategies, such as total quality management and the just-in-time inventory system. Thus, when enterprises achieve sustainable performance through such integrations, their business performance is also indirectly improved. In addition, the implementation of environmental sustainability management practices can reduce a company's environmental risks, enhance innovative technology, and contribute to long-term performance development. Environmental matters affect long-term performance rather than short-term profits or sales [72]. Therefore, a mediation fit model was constructed to show the relationship among enterprise commitment, supply chain cooperation, operation integration, sustainable performance and business performance, as shown in Figure 3. Hypothesis 5 (H5) is proposed as follows:

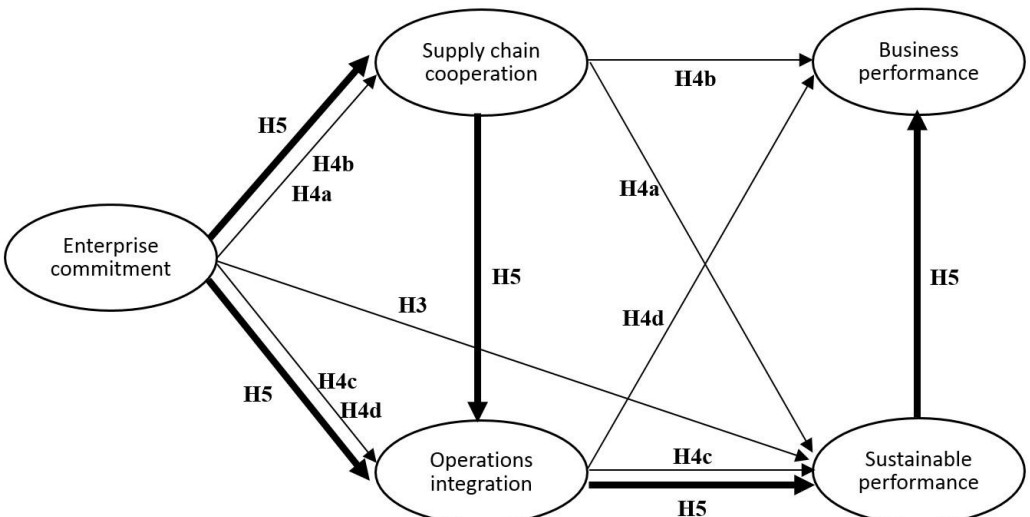

**Figure 3.** Mediation fit model for SSCs.

**Hypothesis 5 (H5).** *Enterprise commitment forms an intermediary mediation fit with supply chain cooperation and operation integration, which has a positive impact on sustainable performance and a subsequent effect on business performance.*

As enterprises shift toward high added value, similar shifts will follow, including increased production design activities or increased service activities. Through effective design and design capabilities, enterprises can improve product quality and launch a comprehensive series of products instead of merely a single product. Furthermore, high-value manufacturers consider brand and reputation more seriously than low-value manufacturers. Customers and reputation are often the driving forces for change, which mostly starts at the design level. Most changes implemented by low-value manufacturers in the upper or middle sector of a supply chain originate from cost factors or competitors, who generally apply changes through manufacturing and logistics. On the bases of the preceding discussion, value-added enterprises are considered to differ substantially from their low-value counterparts. Therefore, Hypothesis 6 (H6) is proposed as follows:

**Hypothesis 6 (H6).** *The supply chain structure adaptation model differs for diverse value-added industries.*

The questionnaire used in this study was composed of two parts. The first part comprised 19 questions involving five dimensions, namely, enterprise commitment, supply chain cooperation, operation integration, sustainable performance, and business performance. The contents of the questionnaire and the references to the questions are presented in Table 2. The second part of the questionnaire was a sample data survey comprising eight questions and divided into three sections, namely, the current state of business operations, the scope of business services provided by enterprises, and the basic information of the respondents (Table 3). The questionnaire employed a seven-point Likert scale to indicate influence. Of the 1705 questionnaires distributed, 290 were returned. Among these, 267 were valid and 23 were invalid, yielding a recovery rate of approximately 15.8%. Subsequently, the assumptions established in this section were tested using SPSS and AMOS 22.

**Table 2.** Questionnaire items concerning factors such as SSC structure and performance.

| Facet | Questionnaire Item |
|---|---|
| A.<br>Enterprise commitment | A1. The importance of "organizational development and growth"<br>A2. The importance of "green industry"<br>A3. The importance of "social responsibility" |
| B.<br>Supply chain cooperation | B1. The maturity of the "production and sales synergy" mechanism<br>B2. The maturity of "customer relationship management"<br>B3. The maturity of the "customer rating mechanism" |
| C.<br>Operations integration | C1. The degree to which market planning is conducted in the context of green products<br>C2. The extent to which production planning is conducted in the context of low-carbon production<br>C3. The extent to which it is conducted in the context of green transport<br>C4. The ability to reduce the "carbon footprint" of a product<br>C5. The extent to which purchases are made in the context of "green suppliers"<br>C6. The extent to which green storage is conducted |
| D.<br>Sustainable performance | D1. The capability to manage" environmentally hazardous substances" (e.g., lead, mercury, and cadmium)<br>D2. The capability to "use chemical substances"<br>D3. The capability to "recycle waste" |
| E.<br>Business performance | E1. The capability to "order delivery"<br>E2. The capability to continually "reduce inventory"<br>E3. The capability to continually "improve product yield"<br>E4. The capability to continually "increase capacity utilization" |

**Table 3.** Questionnaire classifications in the company information.

| Classification | Questionnaire Item |
|---|---|
| Current state of business operations | 1.1. Age of the company<br>1.2. Company patterns<br>1.3. Number of worldwide employees in the company<br>1.4. The global business scale of the company |
| Range of the enterprise's business services | 2.1. The company is in a primary industry<br>2.2. The company's primary business operations are conducted in various countries and regions |
| Respondents' information | 3.1. Respondent's seniority<br>3.2. Respondent's position in the company |

## 4. Results and Discussion

The confidence validity of the proposed model was verified using Pearson's correlation coefficient analysis, confirmatory factor analysis (CFA), differential validity analysis, the SSC structural equation model, and matching path analysis to confirm the degree of model fit and thus verify the proposed hypotheses.

### 4.1. Pearson Correlation Analysis

The Pearson product-moment coefficient is a standardized correlation coefficient that removes the unit from the covariance and examines the multicollinearity problem. Collinear problems with correlation coefficients higher than 0.85 may result in model miscalculations [86]. The results of the Pearson correlation analysis (Table 4) revealed that the correlation between the various facets was lower than 0.85, indicating that the proposed model was acceptable and that it had no multicollinearity problems.

### 4.2. CFA

A measurement model must meet the following conditions to have convergent validity [87]:

1. Factor loads must be higher than 0.7; however, 0.6–0.7 is acceptable under validation analysis. Values lower than 0.6 are deleted;
2. Composite reliability (CR) must be higher than 0.7 but not exceed 0.95;
3. Average variance extracted (AVE) must be higher than 0.5;
4. Square multiple correlations (SMCs) must be higher than 0.5.

**Table 4.** Pearson correlation matrix.

| | Enterprise Commitment | Supply Chain Cooperation | Operations Integration | Sustainable Performance | Business Performance |
|---|---|---|---|---|---|
| Enterprise commitment | 1 | | | | |
| Supply chain cooperation | 0.458 | 1 | | | |
| Operations integration | 0.766 | 0.586 | 1 | | |
| Sustainable performance | 0.609 | 0.475 | 0.680 | 1 | |
| Business performance | 0.383 | 0.654 | 0.374 | 0.419 | 1 |

The factors CR, AVE, and SMCs (Table 5) are examined, which must be compliant with the basic requirements of CFA.

**Table 5.** Confirmatory factory analysis table.

| Facial Surface | Item | Model Parameter Estimation | | | | | Convergence | | |
|---|---|---|---|---|---|---|---|---|---|
| | | Non-Standard Factor Load | S.E. | C.R. | p | Standardization Factor Load | SMC | C.R | AVE |
| Enterprise commitment | A. 1 | 1.000 | | | | 0.834 | 0.461 | | |
| | A. 2 | 1.126 | 0.084 | 13.426 | *** | 0.897 | 0.805 | 0.748 | 0.654 |
| | A. 3 | 0.710 | 0.062 | 11.540 | *** | 0.679 | 0.695 | | |
| Supply chain cooperation | B. 1 | 1.000 | | | | 0.683 | 0.530 | | |
| | B. 2 | 1.304 | 0.127 | 10.247 | *** | 0.914 | 0.836 | 0.747 | 0.611 |
| | B. 3 | 1.043 | 0.100 | 10.457 | *** | 0.728 | 0.466 | | |
| Operations integration | C. 1 | 1.000 | | | | 0.856 | 0.520 | | |
| | C. 2 | 0.817 | 0.062 | 13.191 | *** | 0.713 | 0.547 | | |
| | C. 3 | 0.890 | 0.055 | 16.168 | *** | 0.819 | 0.656 | 0.857 | 0.606 |
| | C. 4 | 0.913 | 0.057 | 15.900 | *** | 0.810 | 0.670 | | |
| | C. 5 | 0.874 | 0.063 | 13.889 | *** | 0.739 | 0.508 | | |
| | C. 6 | 0.827 | 0.062 | 13.410 | *** | 0.721 | 0.734 | | |
| Sustainable performance | D. 1 | 1.000 | | | | 0.757 | 0.593 | | |
| | D. 2 | 1.270 | 0.091 | 13.912 | *** | 0.966 | 0.933 | 0.747 | 0.700 |
| | D. 3 | 1.042 | 0.079 | 13.177 | *** | 0.770 | 0.574 | | |
| Business performance | E. 1 | 1.000 | | | | 0.658 | 0.370 | | |
| | E. 2 | 1.043 | 0.097 | 10.724 | *** | 0.860 | 0.630 | 0.797 | 0.543 |
| | E. 3 | 0.998 | 0.095 | 10.459 | *** | 0.794 | 0.740 | | |
| | E. 4 | 0.830 | 0.097 | 8.509 | *** | 0.608 | 0.433 | | |

*** $p < 0.001$ (highly significant).

### 4.3. Discriminant Validity Analysis

In an effective model, items should only be highly correlated with specific constructs. Therefore, performing differential value analysis requires verifying whether two elements are statistically correlated. Accordingly, AVE is employed in order to compare the related coefficients between AVE and its constructs [88]. The AVE method involves evaluating the index of different constructs and should be lower than the indicator of the same surface. In this study, the AVE value of the diagonal was higher than the square of the standardization correlation coefficient of the lower triangle, indicating that the model had adequate discriminant ability. Table 6 presents the results of the discriminant validity analysis.

**Table 6.** Discriminant validity analysis table.

| | Enterprise Commitment | Supply Chain Cooperation | Operations Integration | Sustainable Performance | Business Performance |
|---|---|---|---|---|---|
| Enterprise commitment | **0.654** | | | | |
| Supply chain cooperation | 0.210 | **0.611** | | | |
| Operations integration | 0.587 | 0.343 | **0.606** | | |
| Sustainable performance | 0.371 | 0.226 | 0.462 | **0.700** | |
| Business performance | 0.147 | 0.428 | 0.140 | 0.176 | 1 |

Note: The values in the diagonal line (bonded font) are the AVE values, which were all larger than the standard correlation coefficient square of the lower triangle.

### 4.4. SEM Analysis of SSCs

Analysis of the verification factors and differential values verifies various components; the results indicate that the proposed model is appropriate. Subsequent SEM analysis verifies the integration model and the hypotheses. In the relevant matching moderation indicator, the chi-squared/degree of freedom should be lower than three. The goodness-of-fit index (GFI) indicates the similarities between the model and the sample data. A value closer to one indicates a higher degree of applicability. In this study, GFI > 0.8 was favorable [89]. Adjusted GFI (AGFI) is a moderate index for adjusting GFI with a

degree of freedom. A value closer to one is desirable; for this study, AGFI > 0.8 was favorable [90]. The comparative fit index (CFI) is the difference between the hypothetical model and the noncovariant-associated independent model. In this study, CFI > 0.9 was favorable. The root mean square error of approximation (RMSEA) lacks moderate indicators; therefore, a smaller RMSEA value is desirable. In this study, RMSEA < 0.8 indicated a favorable degree of modeling. As presented in Figure 4, chi-squared/degrees of freedom = 2.820, CFI = 0.911, GFI = 0.867, AGFI = 0.829, and RMSEA = 0.083. The RMSEA was slightly higher than desired. This extremity was due to the number of samples estimated (300–500 samples) [91].

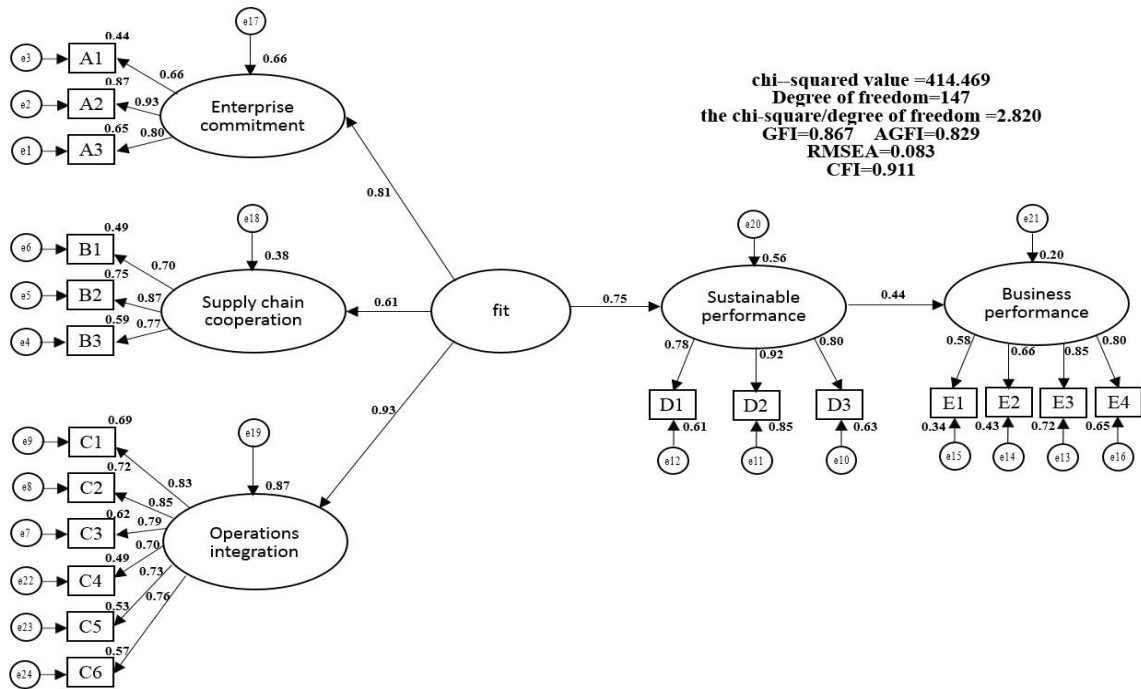

**Figure 4.** SEM analysis diagram.

As presented in Table 7, in terms of the 95% rule, the *p*-value was lower than 0.05, which supported H1 and H2. The level of organizational strategy and corporate operation, including upstream suppliers, internal operational mechanisms, and enterprise commitments, positively influences a company's sustainable performance, which further positively influences business performance.

**Table 7.** Statistical model path coefficient of supply chain.

| Facial Surface | Direction | Facial Surface | Non-Standard Coefficient | Standardization Coefficient | S.E. | C.R. | *p*-Value |
|---|---|---|---|---|---|---|---|
| Sustainable performance | ← | Goodness of fit | 1.053 | 0.749 | 0.116 | 9.071 | *** |
| Enterprise commitment | ← | Goodness of fit | 1.000 | 0.814 | - | - | - |
| Supply chain cooperation | ← | Goodness of fit | 0.630 | 0.614 | 0.084 | 7.463 | *** |
| Operations integration | ← | Goodness of fit | 1.357 | 0.935 | 0.141 | 9.646 | *** |
| Business performance | ← | Sustainable performance | 0.299 | 0.442 | 0.048 | 6.279 | *** |

*** *p* < 0.001 (highly significant)

The results of the covariant standardization adaptation (Figure 4) strongly support the relationships among supply chain cooperation, enterprise commitment, and operating mechanism integration. If the manufacturing industry wishes to strengthen SSCs and improve sustainability and business performance, these three pillars must be improved. In contrast, if supply chain cooperation is low, the other two pillars are likely to be limited as

well. Therefore, path analysis is used to examine other relationships between manufacturers and upstream suppliers.

### 4.5. SSC Path Analysis

Path analysis explores the relationships among enterprise commitment, supply chain cooperation, and operation integration.

#### 4.5.1. Manufacturer Mediation Fit Model: Process Fit

This study examined whether manufacturing has a significant mediation fit and verified whether the hypotheses are valid. The results of the analysis and the path coefficients are presented in Table 8. The results of the analysis validated H3. Only H4b and H4c of H4 were validated. Specifically, operation integration impacts sustainable performance but not business performance. Finally, H5 was assumed to be supported. On the basis of the literature review, this study inferred that sustainable performance can only be generated in products and processes if the knowledge and experience shared among enterprises and their upstream and downstream partners are applied to improve the internal operations of the enterprise. Figure 5 presents the mediation fit process adaptation model, which had a moderate GFI = 0.884, AGFI = 0.846, CFI = 0.928, and RMSEA= 0.076. In Figure 5, significant routes are indicated by bold lines and nonsignificant paths are indicated by dashed lines.

**Table 8.** Mediation fit model–process fit path coefficient table.

| Path | | | Path Coefficient | C.R. | $p$ |
|---|---|---|---|---|---|
| Enterprise commitment | $\rightarrow$ | Supply chain cooperation | 0.465 | 5.965 | *** |
| Enterprise commitment | $\rightarrow$ | Operations integration | 0.628 | 8.110 | *** |
| Supply chain cooperation | $\rightarrow$ | Operations integration | 0.292 | 4.798 | *** |
| Supply chain cooperation | $\rightarrow$ | Sustainable performance | 0.111 | 1.569 | 0.117 |
| Operations integration | $\rightarrow$ | Sustainable performance | 0.456 | 4.325 | *** |
| Enterprise commitment | $\rightarrow$ | Sustainable performance | 0.209 | 2.293 | 0.022 * |
| Supply chain cooperation | $\rightarrow$ | Business performance | 0.612 | 6.227 | *** |
| Operations integration | $\rightarrow$ | Business performance | −0.094 | −0.993 | 0.321 |
| Sustainable performance | $\rightarrow$ | Business performance | 0.214 | 2.465 | 0.014 * |

*** $p < 0.001$ (highly significant); * $p < 0.05$ (significant).

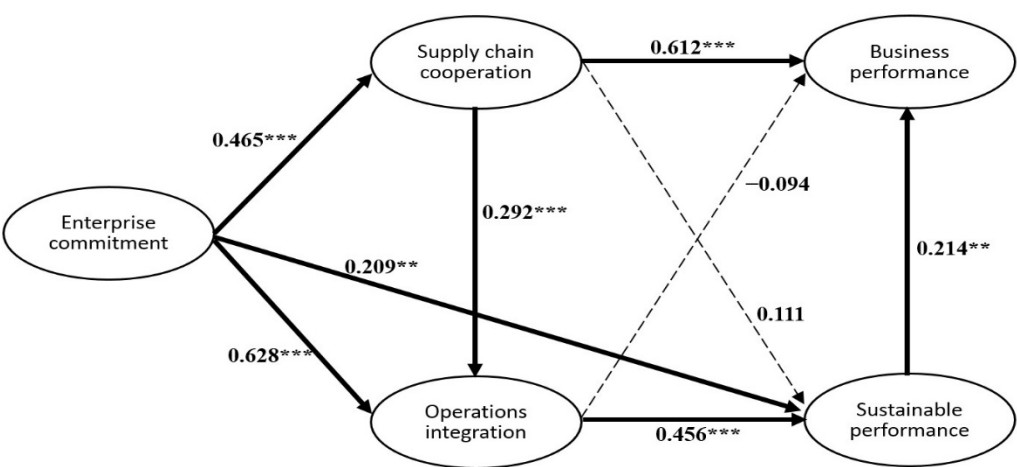

***p < 0.001 (highly significant); **p < 0.01 (significant).

**Figure 5.** Mediation fit: standardized results of the process fit model.

#### 4.5.2. Manufacturer Mediation Fit Model: Industry Value Point

This study classified industries into two groups: those with an added value of ≥25% (high-value manufacturing) and those with an added value of <25% (general manufac-

turing). High-value manufacturing included ready-to-wear apparel, leather, fur and its manufactured goods, wood and bamboo products, printing and data storage, pharmaceuticals, nonmetallic minerals, metal products, electronic components, computers, electronic products, and optical products. In total, 165 and 102 samples were obtained for the high-value and general manufacturing groups, respectively. Path analysis determined whether differences existed in the mediation fit model between enterprises with different added values (Table 9). For both high-value and general manufacturers, commitment had a positive impact on sustainable performance and sustainable growth.

**Table 9.** Mediation fit model–process fit path coefficient table.

| Path | | | High-Value Manufacturing | | General Manufacturing | |
|---|---|---|---|---|---|---|
| | | | Path Coefficient | C.R. | Path Coefficient | C.R. |
| Enterprise commitment | → | Supply chain cooperation | 0.520 | 4.415 *** | 0.489 | 3.861 *** |
| Enterprise commitment | → | Operations integration | 0.595 | 5.248 *** | 0.671 | 5.238 *** |
| Supply chain cooperation) | → | Operations integration | 0.308 | 3.269 ** | 0.277 | 2.951 ** |
| Supply chain cooperation | → | Sustainable performance | 0.093 | 0.943 | 0.126 | 1.029 |
| Operations integration | → | Sustainable performance | 0.486 | 3.373 *** | 0.191 | 1.015 |
| Enterprise commitment | → | Sustainable performance | 0.264 | 2.079 ** | 0.379 | 2.127 ** |
| Supply chain cooperation | → | Business performance | 0.565 | 3.972 *** | 0.622 | 3.856 *** |
| Operations integration | → | Business performance | −0.082 | −0.538 | −0.018 | −0.131 |
| Sustainable performance | → | Business performance | 0.270 | 1.899 | 0.106 | 0.937 |

*** $p < 0.001$ (highly significant); ** $p < 0.01$ (significant).

However, for high-value manufacturing alone, enterprise commitment can act as an intermediary for sustainable performance through supply chain cooperation and operation integration. The positive impact of sustainable performance supports H6. This study also posits that manufacturing industries with high added value are more consistent with this model. Figures 6 and 7 illustrate different value-added enterprises under the mediation fit–process adaptation model. The high-value manufacturing model exhibited a moderate fit (RMSEA = 0.08, GFI = 0.84, AGFI = 0.78, and CFI = 0.91). The general manufacturing path model also had a moderate fit (RMSEA = 0.087, GFI = 0.810, AGFI = 0.748, and CFI = 0.908). Solid lines are significant paths, and dashed lines are nonsignificant paths.

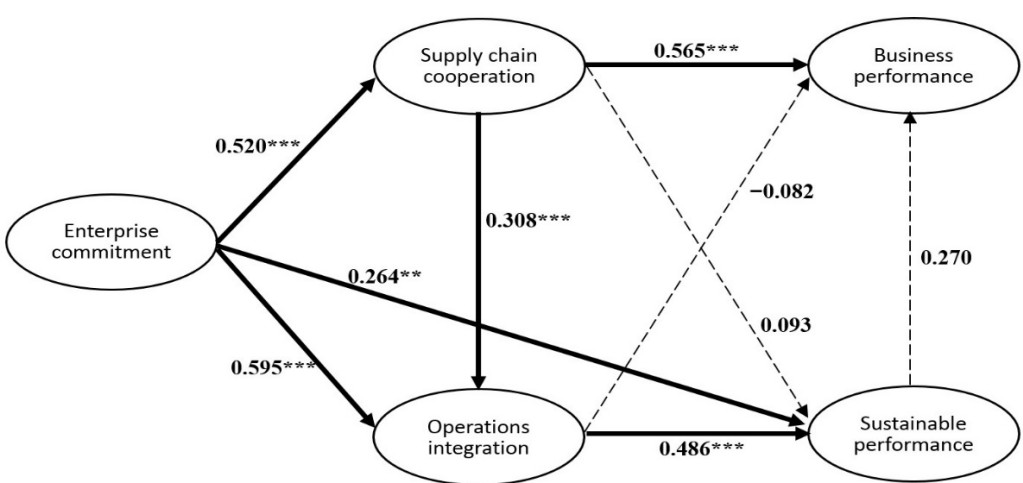

***p < 0.001 (highly significant); **p < 0.01 (significant).

**Figure 6.** Mediation fit: process adaptation model (high-value manufacturing).

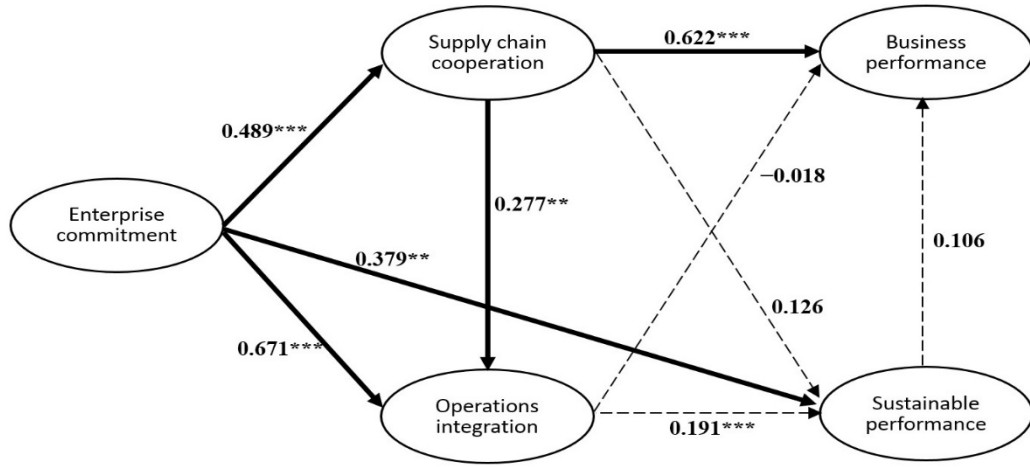

***p < 0.001 (highly significant); **p < 0.01 (significant).

**Figure 7.** Mediation fit: process adaptation model (general manufacturing).

*4.6. Results*

The proposed model comprised three parts: namely, the covariation model, the mediation model, and the industry-oriented intermediary mediation model. These three models were used to test the six hypotheses. The results are presented in Table 10.

**Table 10.** Test results.

|  | **Hypothesis** | ***p*-Value** | **Support or Not** |
|---|---|---|---|
| Covariation Fit Model | H1: In a sustainable manufacturing supply chain, supply chain cooperation, enterprise commitment, and operation integration have a positive impact on sustainable performance. | *** | Support |
|  | H2: Sustainable performance has a positive impact on an enterprise's business performance. | *** | Support |
| Mediation Fit Model | H3: Enterprise commitment has a direct positive impact on sustainable performance. | ** | Support |
|  | H4a: Enterprise commitment has a direct positive impact on sustainable performance through supply chain cooperation. | 0.117 | Not support |
|  | H4b: Enterprise commitment has a direct positive impact on business performance through supply chain cooperation. | *** | Support |
|  | H4c: Enterprise commitment has a positive impact on sustainable performance through operation integration. | *** | Support |
|  | H4d: Enterprise commitment has a positive impact on operating performance through operation integration. | 0.321 | Not support |
|  | H5: Enterprise commitment forms an intermediary mediation fit with supply chain cooperation and operation integration, which has a positive impact on sustainable performance and a subsequent effect on business performance. | ** | Support |
| Mediation Fit Model—industry perspective | H6: The supply chain structure adaptation model differs for diverse value-added industries. | - | Support |

*** *p* < 0.001 (highly significant); ** *p* < 0.01 (significant).

## 5. Conclusions and Future Research

This study developed questionnaires to apply SEM in the context of Taiwan's manufacturing industry concerning how to increase the sustainability of supply chain cooperation and performance through the expansion of sustainable programs or activities in a company's internal operations.

The SSC mechanisms, divided into enterprise commitment, supply chain cooperation, and operation integration and supply chain performance, further identified the level of impact on sustainable performance and business performance. On the basis of this three-faceted approach, firstly, the effects of covariation fit on sustainable performance and business performance were present through the use of SEM; secondly, this study used path analysis to further test the mediation fit model to understand enterprise commitment supply chain cooperation, and operation integration and their impact on performance; and thirdly, this study distinguished high- and low-value industries by using a questionnaire sample analysis to identify differences in the implementation of sustainable practices.

The following results were demonstrated. Firstly, if an internally consistent fitting relationship exists, the implementation of sustainable practices in the context of enterprise commitment, supply chain cooperation, and operation integration has a positive impact on sustainable performance and business performance. Supply chain cooperation is less explanatory than other facets. Secondly, in any case, enterprise commitment has a positive impact on sustainable performance, demonstrating that an enterprise's determination is essential for translating sustainability practices into performance. Thirdly, operation integration affects sustainable performance, and supply chain cooperation affects business performance and sustainable performance. Finally, supply chain cooperation affects sustainable performance through operation integration for high-value industries, but not low-value industries. These results suggest that supply chain cooperation is not the same in industries with different added values.

### 5.1. Academic Implications

An enterprise can improve its sustainability performance and business performance by implementing sustainable practices with utilization resources in a supply chain. In terms of the academic implications, according to resource-based theory [30] and the identified relationships, when an enterprise develops SD as a strategic goal, they must develop the core competence of collaborating with supply chain partners and converting this to the internal process for achieving the goal. Increases in supply chain cooperation can improve business performance, but not sustainable performance, without being associated with an enterprise SD strategy.

In addition, operation integration affects sustainable performance if it involves a sustainability strategy, but it has no significant impact on business performance. This result does indicate a trade-off between sustainability and business performance; internal operations effectively incorporate information shared by supply chain partners. Therefore, sustainable performance can be improved, which contributes to business performance. Thus, SD can satisfy customers and maintain loyalty to sustain the business.

### 5.2. Practical Implications

This study integrated enterprise commitment, supply chain cooperation, and internal mechanisms into a suitable structure and examined their relationship with performance. The relationship between the three pillars and performance was explored through path analysis, which involved the same concept as group contribution architecture in the six orientations of sustainability. Piercy and Brammer [92] suggested that when an organization has a positive influence on the organization in which it operates, it can exhibit excellent performance. However, understanding the interaction through such simplified models alone is challenging.

The motivation for SD in enterprises is the understanding that SD is the only alternative, because all beings and phenomena are mutually interdependent. Every person or organization has a general responsibility to all others [11]. Studying how to use management control tools to make companies more transparent and proactive is recommended; these can then be used to integrate capabilities to support the building of dynamic sustainability capabilities [78]. For example, big data analytics can help identify areas to improve in order to reduce operating costs and accurately forecast demand [93]. Adding value

through sustainable practice can be beneficial, and using the power of the community, can help the industry obtain higher profits.

In terms of management insights, enterprises should consider SD strategies to engage internal and external stakeholders and achieve optimal conditions for sustainable performance. In terms of human factors on organization, senior executives' commitment positively influences on sustainable practices at all stages. Operation integration and sustainable practices can improve sustainable performance, but they will not have an impact on business performance. An enterprise can attempt to quantify sustainable performance to attract customers through building relationships or add value to products. Companies that complete this phase can extend sustainability awareness to the entire supply chain and convert their partners' knowledge into internal sustainably capabilities. Alternating the structure of an enterprise through long-term engagement with supply chain stakeholders ultimately improves business performance. An enterprise which extends these practices to the supply chain will positively influence quantitative operational indicators.

### 5.3. Limitation and Future Research

This study explored only the environmental impacts on sustainable performance. Enterprises also encounter broader social responsibility pressures that are not limited to environmental concerns. In the future, the societal dimension, including employee development and on-site care of network levels, should be incorporated into sustainable performance in order to undertake more comprehensive studies on sustainable performance.

The sample in this study was from Taiwan's manufacturing industry in SME. It is recommended to replicate this methodology for large-scale manufacturing industries, and in supply chains in various countries or regions.

This study revealed, using the adaptation model, that industries with distinct levels of added value have different effects on SSC structures. In the future, it is suggested that enterprises in the same value chain can be rejoined in terms of their midstream and downstream differences as an extension of the industry's added value. This will provide a comprehensive structure between value-added enterprises and SSCs.

**Author Contributions:** Conceptualization, J.-D.L. and C.-C.H.; methodology, J.-D.L.; software, Y.-W.H.; validation, J.-D.L., L.J.-H.L. and Y.-W.H.; investigation, C.-C.H.; data curation, J.-D.L.; writing—original draft preparation, Y.-W.H.; writing—review and editing, J.-D.L.; visualization, Y.-W.H.; supervision, L.J.-H.L.; project administration, J.-D.L.; funding acquisition, J.-D.L. All authors have read and agreed to the published version of the manuscript.

**Funding:** This paper was funded by the Ministry of Science and Technology, Taiwan.

**Institutional Review Board Statement:** Not applicable.

**Informed Consent Statement:** Not applicable.

**Acknowledgments:** This research was supported by the project "A study on the supply chain feature of small and medium-sized enterprise in Taiwan: Perspective of fit", sponsored by the Ministry of Science and Technology, Taiwan, under grant no. MOST108-2410-H-008-060.

**Conflicts of Interest:** The authors declare no conflict of interest.

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
