# Peer review of "Sustainable Supply Chains: Evidence from Small and Medium-Sized Manufacturers"

_sustainability, doi:10.3390/su13169059_

Round 1

Reviewer 1 Report

I checked both versions of the paper (the previous and  the current) and I can tell that all my comments were taken into account during imroving the text. The paper is improved in propre way, so I think it can be published after checking English language.

Author Response

Dear reviewer, Please see the attachment. Thanks !

Reviewer 2 Report

The manuscript had been improved  but still need english corrections

Author Response

Dear reviewer, Please see the attachment. Thank you very much for your efforts. 

Reviewer 3 Report

A sustainable supply chain model for small and medium-sized manufacturers is studied in this current study. The authors make a good revision. However, the article has different drawbacks based on novelty, writing issues, structure, etc. Thus, a major revision is required for considering further. My suggestions are as follows:

  1. Major novelty and findings should be clearly presented in the Abstract section. The current abstract fails to provide exact novelty. Thus, carefully revised the abstract section. Keywords should be abbreviation-free.
  2. The introduction section fails to describe the necessity of your study. Thus rewrite the Introduction section along with proper novelty and research gaps.
  3. Make an author(s) contribution table to show the novelty and research gaps. (For instance, see “Autonomation policy to control work-in-process inventory in a smart production system, International Journal of Production Research, 2021.”)
  4. A graphical representation is required to show the novelty of your study.
  5. The conclusion section should be rewritten in a proper scientific way. More future extension is needed along with proper references.
  6. Remove all “I”, “We”, “our” throughout the manuscript and rewrite the sentence.
  7. Add more current references from 2019,2020, and 2021 and some references from this journal.

Author Response

Dear reviewer, Please see the attachment. Thank you very much ! 

Reviewer 4 Report

I thank the authors for revising and resubmitting the manuscript.

The authors have made numerous changes that make it much easier to understand the text. Certainly, paragraphs 4 and 5 have benefited greatly from the cuts made and the reorganization. Now even the objectives are much clearer, and the motivations that led to cite some theoretical frameworks.

I suggest the authors verify the citations' correctness, particularly regarding the text reported in quotation marks but without bibliographic reference (e.g. lines 133-137).

Round 2

Reviewer 3 Report

The authors made a good revision. However, I only suggest that the future extensions should be properly referenced.

This manuscript is a resubmission of an earlier submission. The following is a list of the peer review reports and author responses from that submission.

Round 1

Reviewer 1 Report

The paper is very intresting. It deals with the subject in a very interesting way. I can see that the authors put a lot of work into research and preparing this paper. I have some commets, but they do not lower the scientific value of this paper. However, it should be corrected.

Main comments:

  • Page 1-2. Introduction is ok. I have small problem with one thing – the purpose of the paper is not clearly presented. It should be underlined for reads. (I ticked "Are the research design, questions, hypotheses and methods clearly stated?" can be improved)
  • Page 7, Figure 1. I found two problems: there is nothing about the Figure in the text (you should mentioned about it in the text). Secondly, there is a mistake in Figure SUSTAINABLE PRFORMACE – it should be PeRFORMACE
  • Page 9, Figure 2 - there is nothing about the Figure in the text
  • Page 10, line 425: You are writing that 1,705 questionnaires distributed, 290 were returned. So you have only 290 results. I think you should change in the abstract, because 290 companies were surveyed, you do not have results from the rest.
  • Page 10, table 2: First two lines are the same. Are both needed?
  • Page 11, chapter 4.1. It is similar to Figure 1&2. I do not see the results. I saw them in the next part of the text (Table 3), but you should mention something about this table in this chapter. Lack of mentioning about table or figures in the text is very misleading. Maybe you can move the table to this chapter.
  • Page 14, table 7. The title of the table should be above the table, not below.
  • Pages 15-16. Figures 4-6. It is the same mistake as in Figure 1: SUSTAINABLE PRFORMACE – it should be PeRFORMACE
  • Page 17, line 576. You have Iimplementation. You should remove “i” (the second letter)

Reviewer 2 Report

The manuscript proposed a model to understand sustainable supply chains relationships by

incorporating enterprise commitment, supply chain integration and operational integration approach and analyses the relationship between sustainable performance and business performance. The results show that in long-term sustainable practices influence in a positive way the operational quantitative indicators.The study is a step forward in supply chain performance research and it is a contribution to enterprises implementation of sustainable practices. The methodology is clearly explained   and theoretically grounded although some minor corrections needs to be done: from line 93 to 100, the statements should be supported by references. Table 1 and table 2 have the same title. Change table 2 title in accordance with the contents. Table 9: in H1, must be changed “en-terprise” by “enterprise”. In H4b, Change “perfor-mance” by “performance”. Size of Column 4 must be adjusted. Line 591, change “high- and” by “high and”.

Reviewer 3 Report

The same paper with the same hypothesis already published in this journal. Nothing no new contribution in the current study. Thus, commenting on an already published paper is worthless.

Reference

Kot, S. 2018. Sustainable Supply Chain Management in Small and Medium Enterprises, Sustainability, 10,1143.

Reviewer 4 Report

Please consult the attached file
